# Self-care knowledge, attitude and associated factors among outpatients with diabetes mellitus in Arsi Zone, Southeast Ethiopia

**Rahel Nega Kassa**[1]*, **Hana Abera Hailemariam**[2], **Mekdes Hailegebreal Habte**[3], **Altayework Mekonnen Gebresillassie**[1]

1 Department of Surgical Nursing, Saint Paul Millennium Medial College, Addis Ababa, Ethiopia,
2 Department of Neonatal Nursing, Saint Paul Millennium Medial College, Addis Ababa, Ethiopia,
3 Department of Pediatrics Nursing, Saint Paul Millennium Medial College, Addis Ababa, Ethiopia

* rahelnega208@gmail.com, rahel.nega@sphmmc.edu.et

**Data Availability Statement:** All the relevant data are within the manuscript itself.

## Abstract

### Introduction

Diabetes mellitus is a chronic illness that requires continuing medical care and ongoing patient self-management, education and support to prevent acute complications and to reduce the risk of long-term complications. Therefore, this study aims to assess the self-care knowledge, attitude and associated factors among outpatients with diabetes mellitus in Arsi Zone, Southeast Ethiopia.

### Materials and methods

A cross sectional study was employed in Arsi Zone, Southeast Ethiopia from April to June 2017 among 290 patients with diabetes mellitus. Structured questionnaire was employed through face to face interview. Bivariate and multivariate regression was done and a statistical significance was declared at p value < 0.05.

### Results

Among 290 respondents, 64.8% and 27.6% of them had good knowledge and good attitude towards self-care in this study respectively. Being married (AOR: 3.41, 95% CI: 1.480–8.095), being employed in occupation (AOR: 5.8, 95% CI: 2.26–14.67) and living in higher socioeconomic status (AOR: 2.0, 95% CI: 1.096–3.322) are the independent factors associated to good knowledge of respondents towards self-care whereas living in lower socioeconomic status(AOR: 0.478, 95% CI: 0.262–0.874), having informal education (AOR: 4.002, 95% CI: 1.941–8.254), not having family history of diabetes mellitus (AOR: 0.422, 95% CI: 0.222–0.803) and having short duration of diagnosis (AOR: 3.209, 95% CI: 1.537–6.779) were significantly associated factors to have poor attitude towards self-care.

### Conclusion

Majority of the study participants had good knowledge towards diabetes self-care practice while a significant number of the participants had unfavorable attitude towards diabetes self-

**Funding:** The study was funded by Arsi University. The funders had no role in study design, data collection and analysis, decision to publish, or preparation of the manuscript. The views presented in the article are not necessarily expressing the views of the funding organization.

**Competing interests:** The authors have declared that no competing interest exists.

**Abbreviations:** AOR, adjusted odds ratio; BMI, body mass index; CI, confidence interval; COR, crude odds ratio; DAS3, Diabetes Attitude Survey; DM, diabetes mellitus; Epi info, Statistical Package for Epidemiological Information Analysis; FBS, Fasting Blood Sugar; HMIS, Health Management and Information System; MCCs, Multiple chronic conditions; MPH, Master of Public Health; MSc., Master of Science in Nursing; SDSCA, Summary of Diabetic Self-Care Activities; SKILLDS, Spoken Knowledge In Low Literacy Diabetes Knowledge Assessment Scale.

care. Being married, being employed and living in higher socioeconomic status were the determinant factors of knowledge towards the diabetes self-care practice while having informal education and having short duration of diagnosis were the significant factors associated to the unfavorable attitude towards diabetes self-care practice. Therefore, efforts should be made on enhancing patients' socioeconomic status and equipping with diabetic self-care centered health information particularly for those patients with short duration of diagnosis.

## Introduction

Chronic health conditions are responsible for 60% of the global disease burden. Globally, approximately one in three of all adults suffer from multiple chronic conditions (MCCs) [1]. In this group of health conditions, diabetes mellitus stands out because of high morbidity and mortality rates, as well as increasing prevalence levels [2]. About 422 million people worldwide have diabetes, the majority living in low-and middle-income countries, and 1.6 million deaths are directly attributed to diabetes each year [3]. In developing countries, treatment adherence reaches a mere 20%, generating negative health statistics and entailing high costs for families, society and governments [4] Thus, integral care for diabetes patients should cover psychosocial and cultural aspects. Therapeutic education is fundamental to inform, motivate and strengthen patients and families to live with the chronic condition [5].

In Ethiopia, one of the top five African countries with high prevalence of diabetes in the age range of 18–99 years, there were 2,652,129 cases of diabetes in 2017 [6]. A cross sectional study conducted in Ethiopia showed that almost half of the participants (n = 320) (49%) had long term diabetic complications confirmed medically [7]. Studies revealed that the diabetic care in general is below the acceptable standard [8]. Though national data on prevalence and incidence of diabetes are lacking in Ethiopia, patient attendance rates and medical admissions in major hospitals are rising. Inadequate diabetic self-management remains a significant problem facing health care providers and populations in all settings. In contrast, patients who have adequate self-management have better outcomes, live longer, enjoy a higher quality of life, and suffer fewer symptoms and minimal complications [9]. So that, Diabetes Self-Management is the cornerstone of care for all individuals with diabetes who want to achieve successful health-related outcomes [10]. There is an increasing amount of evidence that individuals who are educated and diligent in their diabetes self-care achieve better and durable diabetic control [11,12]. Past studies on knowledge, attitude, and practice towards the prevention of diabetes complications consistently revealed the requirements of better awareness on prevention, diagnosis, and risk factor control of diabetes [13]. A good attitude towards DM complications helps patients to change any harmful dietary and lifestyle habits [14].

Though general health education regarding diabetes mellitus self-care activities and its complication is being provided by health care providers working in different health settings, well organized diabetic self care education program is not yet established in Ethiopian health institutions. As to the investigators knowledge, there is no study conducted on this topic in the present study area. So, this study intended to assess self-care knowledge, attitude and associated factors among diabetic outpatients in Arsi Zone, Southeast Ethiopia.

## Materials and methods

A total of 290 adult study participants were included in this study. The study was conducted from April to June 2017. Data were collected by using pretested and structured questionnaire

through face to face interview. The questionnaire was developed after reviewing relevant literature related with the problem under the study [15,16].

## Study design

A cross-sectional study design was employed.

## Study area

Facility based study which was performed among patients with diabetes who had follow up at four hospitals (AsellaReferral Hospital, Bekoji Hospital, Abomssa Hospital and Arsi Robe Hospital) found in Arsi Zone, Ethiopia.

**Study period:** April to June, 2017.

**Sample size:** 290 patients were recruited.

**Sample size calculation:** The study used the single population proportion sample size determination formula. The prevalence rate of good attitude towards diabetic self-care practice was taken as 78% from a study conducted at Dilla UniversityReferral Hospital ([9]), with 95% CI, and 5% marginal error (where n is desired sample size, Z is value of standard normal variable at 95% confidence interval and, p is maximum expected proportion which is 78% and d is marginal error which is 5%).

n = Z 2 α/2 P (1-P)/d2 = (1.96)2* 0.78* 0.22 / (0.05)2

= 264

Adding 10% contingency for non-response, the final total sample size was 290.

## Sampling technique

The monthly flow of the patients was determined from the Health Management and Information System (HMIS) of each hospital found in Arsi Zone. The total monthly number of the patients obtained from those four hospitals found in Arsi Zone was 507, 371, 322, and 302 respectively. The number of study participants was proportionally allocated for each hospital to maintain the representativeness of the data (99, 71, 62, and 58 respectively).Every fifth patient was selected using the systematic sampling technique for interview. Simple random sampling or lottery method was used to select the first patient to start the interview.

## Inclusion criteria

All patients with DM aged 18 years and older who were on follow up at least for the past six months prior to the study at Hospitals found in Arsi Zone.

## Exclusion criteria

Diabetes patients who were, severely ill, mentally and physically incapable for the interview.

## Operational definitions

**Knowledge:** study participants who score greater than 50% were considered to have "good knowledge" and those who score 50% or less were considered to have "poor knowledge".

**Attitude:** study participants who score 50% and above were considered as having a "good attitude" and respondents who score less than 50% considered as having "poor attitude" for self-care.

### Data collection tool and technique

Data was collected by using pretested and structured questionnaire through face to face interview. The tool was adapted from SKILLDS (Spoken Knowledge In Low Literacy Diabetes Knowledge Assessment scale) [15] and DAS$_3$ (Diabetes Attitude Survey) [16]. It contained socio-demographic characteristics of the respondents; clinical characteristics of the participants like type of diabetes mellitus, duration of diagnosis, presence of complications and co-morbidities were developed from different literatures. The data related to these complications and comorbidities were collected from each patient's medical chart. Afaan Oromo language version of questionnaire was used for data collection purpose. Besides, for Amharic language only speaker study subject, in order to avoid translation bias at the spot, each data collector had one Amharic language questionnaire that was used as a dictionary.

Twelve health professionals who were fluent in speaking Amharic and Afaan Oromo were involved in the data collection process. Eight Masters of Public Health (MPH) and Masters of Science in Nursing (MSc) holder health professionals were recruited as supervisors.

### Data quality assurance

The questionnaire was translated from English language to Amharic and Afaan Oromo by different translator and back to English by second other translator who was a health professional and fluent on the respective languages to compare its consistency.

The questionnaire was pretested to determine its validity. It was pretested on 5% of the total sample size in patients on follow up at Adama Hospital and necessary adjustments was made on the questionnaire before it was used for actual data collection. Data collectors and supervisors were trained for three days on the study instrument and data collection procedure. The principal investigator and the supervisors checked the collected data for completeness and corrective measures were taken accordingly.

## Statistical analysis

The data was checked for completeness and consistencies, then, it was cleaned, coded and entered in to computer using SPSS windows version 21.

Descriptive statistics was computed to describe the socio-demographic characteristics of the study participants and determine the diabetic self-care practice knowledge and attitude of the study participants. Additionally, binary and multiple logistic regression analyses were constructed to examine the existence of relationship between the diabetic self-care practice knowledge and attitude and selected variables. Statistical significance was declared at P<0.05. Finally, the result was presented in the form of text and using tables.

### Ethics approval and consent to participate

The approval letter was taken from the Ethical Review of College of Health Sciences of Arsi University (Ref. no: CHS/R/0019/2016/17). The final permission letter was written to the respective hospitals by the ethical review committee of Arsi Zone Health Bureau. The will of the participants to participate in the study was assured by informed written consent, which was obtained from each study participant before data collection begins. The consent was taken both for the interview and to take some clinical information from their medical records. The right and the confidentiality of study participants were assured.

## Result

### Socio-demographic characteristics of respondents

All the sampled study participants were involved in this study giving a response rate of 100 percent. The mean age of respondents was 39 ± 15.8. Most of the respondents 63 (21.1%) were within 18–24 age group and majority of the respondents 142 (47.5%) were Muslim in religion. Of these 290 respondents, 79 (36.6%) had attained elementary school and above. Majority of the study participants, 202(67.6%) were married, and 177(60.1%) of them are poor in their socioeconomic status. Large number 158 (53%) of the study participants were farmers followed by private business holders 54(19.2%). In addition, majority 166(55.5%) of the respondents were from the rural area (Table 1).

### Clinical characteristics of the respondents

Majority 198 (66.2%) of the respondents had normal body weight (18.5–24.9kg/m2). More than half, (58.3%), of the respondents had type 1 diabetes. Most of the respondents, 190 (65.5%) were using insulin whereas only 4.8%of them were taking both insulin and oral

**Table 1. Socio-demographic characteristics of patients with diabetes mellitus on follow up at hospitals of Arsi Zone, Southeast Ethiopia, 2017 (n = 290).**

| Variables | Category | Frequency | Percent |
|---|---|---|---|
| **Sex** | Male | 176 | 60.7 |
| | Female | 114 | 39.3 |
| **Age** | 18–24 | 63 | 21.1 |
| | 25–34 | 70 | 23.4 |
| | 35–44 | 56 | 18.7 |
| | 45–54 | 48 | 16.1 |
| | 55–64 | 37 | 12.4 |
| | >65 | 25 | 8.4 |
| **Religion** | Muslim | 138 | 47.6 |
| | Orthodox Christian | 128 | 44.1 |
| | Protestant | 21 | 7.2 |
| | Catholic | 3 | 1 |
| **Educational status** | Cannot read and write | 65 | 21.7 |
| | Can read and write | 36 | 12.4 |
| | Elementary | 40 | 13.8 |
| | High school | 19 | 6.6 |
| | College and above | 130 | 44.8 |
| **Marital status** | Single | 85 | 29.3 |
| | Married | 194 | 66.9 |
| | Divorced | 7 | 2.4 |
| | Widowed | 4 | 1.4 |
| **Socioeconomic status** | Poor | 154 | 53.1 |
| | Medium | 41 | 14.1 |
| | Rich | 95 | 32.8 |
| **Occupation** | Farmer | 156 | 53.8 |
| | Private business | 54 | 18.6 |
| | Student | 45 | 15.5 |
| | Employed | 35 | 12.1 |
| **Place of residence** | Urban | 130 | 44.8 |
| | Rural | 160 | 55.2 |

**Table 2. Clinical characteristics of patients with diabetes mellitus follow up at hospitals of Arsi Zone, Southeast Ethiopia, 2017 (n = 290).**

| Variables | Category | Frequency | Percent |
|---|---|---|---|
| Type DM | Type 1 | 169 | 58.3 |
| | Type 2 | 121 | 41.7 |
| Family history | Yes | 61 | 21 |
| | No | 229 | 79 |
| Type of treatment | Insulin Injection | 190 | 65.5 |
| | Oral anti hyperglycemic | 86 | 29.7 |
| | Both | 14 | 4.8 |
| BMI(kg/m$^2$) | Underweight (<18.5) | 49 | 16.9 |
| | Normal weight1 (18.5–24.9) | 190 | 65.5 |
| | Overweight (25–29.9) | 47 | 16.2 |
| | Obese (30 and above) | 4 | 1.4 |
| Having glucometer | Yes | 40 | 13.8 |
| | No | 250 | 86.2 |
| Diabetic complications | Yes | 72 | 24.8 |
| | No | 218 | 75.2 |
| Presence of comorbidities | Yes | 79 | 27.2 |
| | No | 211 | 72.8 |
| Duration of diagnosis(in years) | <5 | 97 | 33.4 |
| | 5–10 | 93 | 32.1 |
| | >10 | 100 | 34.5 |

hypoglycemic agents. Most (72%) of the respondents had no diabetic complications and 72.8% of the respondents had no co morbidities (Table 2).

## Factors associated to knowledge of self-care practice among patients with diabetes mellitus

Above half (51.2%) of the study participants had good knowledge in this study. Marital status, occupation, andsocio economic status were found to be a significant associated factors affecting diabetes self-care knowledge of the study participants. Those respondents who had married were 3.41 times more likely to had good diabetes self-care knowledge than the single ones (AOR: 3.41, 95% CI: 1.480–8.095). On the other hand, study participants who had employed were 5.8 times more likely to have good diabetes self-care knowledge than farmers (AOR: 5.8, 95% CI: 2.26–14.67). Those respondents who were living in higher socioeconomic status were 2 times more likely to had good diabetes self-care knowledge than those who were living in lower socioeconomic status (AOR: 2.0, 95% CI: 1.096–3.322) (Table 3).

## Factors associated to attitude towards self-care among patients with diabetes mellitus

Among the respondents, about 219(73.2%) of them had poor attitude towards diabetes self-care activities. Monthly income, educational status family history and duration of diagnosis were found to be substantial factors associated with diabetes self-care attitude of the respondents. Accordingly, those respondents who were living in lower socioeconomic status were 47.8% more likely to had poor attitude towards diabetes self-care than those who are living in higher socioeconomic status (AOR: 0.478, 95% CI: 0.262–0.874). In other way, study participants who had informal education were 4.002 times more likely to have poor diabetes self-care attitude than study participants who had college and above education (AOR: 4.002, 95% CI:

**Table 3. Factors associated to knowledge of self-care among diabetes mellitus outpatients in Arsi zone, Southeast Ethiopia (n = 290).**

| Variables | Self-care knowledge status | | COR (CI 95%) | AOR (CI 95%) | P value |
|---|---|---|---|---|---|
| | Good N% | Poor N% | | | |
| **Marital status** | | | | | |
| Single | 47(55.3) | 38(44.7) | 1 | 1 | |
| Married | 135(69.6) | 59(30.4) | 2.039(1.217–3.416) | 3.461(1.480–8.095) | 0.004** |
| Divorced | 5(71.4) | 2(28.6) | 4.015(0.736–21.901) | 5.964(0.917–38.777) | 0.062 |
| Widowed | 1(25) | 3(75) | 1.606(0.216–11.957) | 3.961(0.408–38.408) | 0.235 |
| **Occupation** | | | | | |
| Farmer | 95(60.9) | 61(39.1) | 1 | 1 | |
| Private business | 33(61.1) | 21(38.9) | 2.3(1.223–4.324) | 3.322(1.609–6.858) | 0.001** |
| Student | 28(62.2) | 17(37.8) | 1.011(0.522–1.957) | 2.880(1.121–7.403) | 0.028** |
| Employed | 32(91.6) | 3(8.6) | 4.6(1.976–10.709) | 5.766(2.266–14.670) | 0.000** |
| **Monthly income** | | | | | |
| Poor | 97(63) | 57(37) | 1 | 1 | |
| Medium | 23(56.1) | 18(43.9) | 0.686(0.433–1.736) | 0.912(0.419–1.985) | 0.817 |
| Rich | 68(71.6) | 27(28.4) | 0.475(0.273–0.826) | 2(1.09–3.322) | 0.022** |

** shows statistically significant association.

1.941–8.254). The other finding was that study participants who had not family history of diabetes mellitus had 42.2% times more likely to have poor attitude towards diabetes self-care compared to those respondents who had family history of diabetes mellitus (AOR: 0.422, 95% CI: 0.222–0.803). On the other hand, study participants who had short duration of diagnosis were 3.209 times more likely to have poor attitude than who had long duration of diagnosis (AOR: 3.209, 95% CI: 1.537–6.779) (Table 4).

**Table 4. Associated factors of attitude towards self-care among diabetes mellitus outpatients in Arsi Zone, Southeast Ethiopia (n = 290).**

| Variable | Self-care attitude status | | COR (CI 95%) | AOR (CI 95%) | P value |
|---|---|---|---|---|---|
| | Good N% | Poor N% | | | |
| **Monthly Income** | | | | | |
| Poor | 32(20.8) | 122(79.2) | 0.486(0.275–0.857) | 0.478(0.262–0.874) | 0.017** |
| Medium | 27(65.85) | 14(34.15) | 0.956(0.445–2.052) | 0.844(0.372–1.915) | 0.684 |
| Rich | 61(64.21) | 34(35.79) | 1 | 1 | |
| **Educational status** | | | | | |
| Cannot read and write | 25(69.4) | 11(30.6) | 1.966(0.855–4.521) | 2.750(0.734–4.172) | 0.207 |
| Can read and write | 39(60) | 26(40) | 3.217(1.648–6.283) | 4.002(1.941–8.254) | 0.000** |
| Elementary | 25(62.5) | 15(37.5) | 2.784(1.279–6.062) | 2.866(1.255–6.546) | 0.012** |
| High school | 14(73.7) | 5(26.3) | 1.508(0.502–4.531) | 1.520(0.492–4.693) | 0.466 |
| College and above | 107(82.3) | 23(17.7) | 1 | 1 | |
| **Family history** | | | | | |
| Yes | 36(59) | 25(41) | 1 | 1 | |
| No | 174(76) | 55(24) | 0.433(0.239–0.783) | 0.422(0.222–0.803) | 0.017** |
| **Duration of diagnosis(in Years** | | | | | |
| <5 | 67(69.1) | 30(30.9) | 1.774(0.921–3.416) | 2.347(1.152–4.779) | 0.019** |
| 5–10 | 62(66.7) | 31(33.3) | 2.116(1.096–4.084) | 3.209(1.537–6.700) | 0.002** |
| >10 | 81(81) | 19(19) | 1 | 1 | |

** shows statistically significant association.

## Discussion

This study tried to assess self-care knowledge and attitude among patients with diabetes in Arsi Zone and 64.9% [95% CI (59.0–70.2%)], 26.8% [95% CI (22.8–33.7)] of the respondents had good knowledge and good attitude towards self-care respectively.

Regarding knowledge, the finding of this study is comparative with the study conducted at North Shewa Zone Oromia, Ethiopia (67.8%) and Egypt (52.3%) [17,18].

However, it is lower than the study done at Ayder Hospital Tigray and in Nigeria [19,20]. The variation from the study conducted at Ayder Hospital might be that almost 79% of the study participants at Ayder Hospital were from urban area that possibly gives them the chance of getting more information than the respondents from the present study area where 55.2% of the respondents were from rural area.

Besides, almost fifty percent (49.5%) of the study participants included under the study conducted at Nigeria had taken tertiary level education whereas only 17.7% of the study participants in the present study were attended the tertiary level or college and above. This might contribute to difference in gaining self-care related knowledge from variety of sources.

Being married, being employed and living higher socioeconomic status were significantly associated factors to had good self-care knowledge. This can be explained as married ones might have chance of detail discussion with their couple and information exchange about the disease. In addition to this support to access the awareness from different Medias might be available. Similarly, those respondents who were employed have possibility of getting aware-ness from different directions since they can have communication and information exchange from variety of individuals or professionals. In addition, study participants who are living in higher socioeconomic status might have probability of accessing medias as well as health professionals than those who are living in the lower level of economic status.

On the other hand, 26.8% of the respondents in this study had good attitude towards diabe-tes mellitus self-care which is far less from the study conducted in Debre Tabor Town, North-west Ethiopia (39.5%) [21], Kenya (49%) [22], Adama, Ethiopia (81.9%) [23] where respondents had positive attitude towards diabetes self-care management. This variation might be due to the difference in study population, where majority of participants from Debre Tabor were civil servants (62.1%), exposed to health information (50.4%) unlike the partici-pants in this study where majority of them were from rural area (55.2%), under poor socio eco-nomic status (53.1%), having no family history of DM (79%), were farmers (53.8%) and also similar in Kenya. This reveals that living in lower socioeconomic status, having informal edu-cation, not having family history of diabetes mellitus were significantly associated with having poor attitude towards self-care on diabetes mellitus.

The finding that is revealed in Bale Zone, Ethiopia [24] also confirmed that respondents in higher economic status had good attitude than in the lower economic status.

In this study, participants who had short duration of diagnosis were 3.209 times more likely to have poor attitude than who had long duration of diagnosis. This finding coincides with the study done in Brazil [25] which shows that as Diabetes is a chronic disease, the shorter the tim-ing of the diagnosis, the more conflict feelings that needs to overcome in order to reach the stage of positive coping of the disease that in turn affects the attitude.

The present study revealed that most of the respondents had unfavorable attitude though most of them had good knowledge. This could be due to the fact that good knowledge does not necessarily guarantee that the patient will have good attitude. As different scientific researches [26] show people who have better knowledge are more resistance and questionable rather than implementing what they know rather than those who know less. Knowledgeable

members of the public are in general, less supportive of morally contentious areas than those who are less knowledgeable.

## Conclusion

Majority of the study participants had good knowledge towards self-care in this study. Therefore, it has to be strengthening giving continual diabetic self-care centered health education for patients who have follow up on those respective hospitals found in Arsi Zone. On the contrary, many of the respondents had poor attitude towards self-care so that efforts should be made from different stakeholders like hospitals, health professionals who are following cases these patients. This can be in terms of delivering clear and brief awareness which can in turn result in change in behavior towards self-care, supporting those who are living in lower socio economic status.

Therefore, efforts should be made on equipping patients with adequate and specific diabetic self-care centered health information particularly for those patients with short duration of diagnosis. In other way enhancing the socioeconomic status of the patients is highly recommended.

## Acknowledgments

We would like to thank Arsi University, School of Health Sciences and Arsi Zone Health Bureau, for the support of necessary materials while doing this study. We also like to thank the directors of the hospitals, data collectors, and study participants.

## Author Contributions

**Conceptualization:** Rahel Nega Kassa.

**Data curation:** Rahel Nega Kassa, Hana Abera Hailemariam, Mekdes Hailegebreal Habte.

**Formal analysis:** Rahel Nega Kassa.

**Funding acquisition:** Rahel Nega Kassa.

**Investigation:** Rahel Nega Kassa.

**Methodology:** Rahel Nega Kassa, Hana Abera Hailemariam, Altayework Mekonnen Gebresillassie.

**Project administration:** Rahel Nega Kassa.

**Resources:** Rahel Nega Kassa.

**Software:** Rahel Nega Kassa, Altayework Mekonnen Gebresillassie.

**Supervision:** Rahel Nega Kassa, Hana Abera Hailemariam, Mekdes Hailegebreal Habte, Altayework Mekonnen Gebresillassie.

**Validation:** Rahel Nega Kassa, Mekdes Hailegebreal Habte, Altayework Mekonnen Gebresillassie.

**Visualization:** Rahel Nega Kassa.

**Writing – original draft:** Rahel Nega Kassa.

**Writing – review & editing:** Rahel Nega Kassa, Hana Abera Hailemariam, Mekdes Hailegebreal Habte, Altayework Mekonnen Gebresillassie.

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
