## [Decision Letter · Decision Letter 0]

4 Aug 2021

 PGPH-D-21-00297 Self-Care Knowledge, Attitude and Associated Factors among Outpatients with diabetes mellitus in Arsi Zone, Southeast Ethiopia PLOS Global Public Health

Dear Dr. %Kassa%,

Thank you for submitting your manuscript to PLOS Global Public Health. After careful consideration, we feel that it has merit but does not fully meet PLOS Global Public Health’s publication criteria as it currently stands. Therefore, we invite you to submit a revised version of the manuscript that addresses the points raised during the review process.

 EDITOR: Please insert comments here and delete this placeholder text when finished. Be sure to: Indicate which changes you require for acceptance versus which changes you recommendAddress any conflicts between the reviews so that it's clear which advice the authors should followProvide specific feedback from your evaluation of the manuscript  

Please ensure that your decision is justified on PLOS Global Public Health’s publication criteria and not, for example, on novelty or perceived impact.

We look forward to receiving your revised manuscript.

Kind regards,

Palash Chandra Banik, MPhil

Academic Editor

Journal Requirements:

Additional Editor Comments (if provided):

- Please consider incorporating the comments of the both reviewers. Beside this I would like to request for the possible language editing.

Reviewers' comments:

Reviewer's Responses to Questions

**Comments to the Author**

1. Does this manuscript meet PLOS Global Public Health’s publication criteria? Is the manuscript technically sound, and do the data support the conclusions? The manuscript must describe methodologically and ethically rigorous research with conclusions that are appropriately drawn based on the data presented.

Reviewer #1: Partly

Reviewer #2: Yes

2. Has the statistical analysis been performed appropriately and rigorously?

Reviewer #1: No

Reviewer #2: Yes

3. Have the authors made all data underlying the findings in their manuscript fully available (please refer to the Data Availability Statement at the start of the manuscript PDF file)?

Reviewer #1: Yes

Reviewer #2: No

4. Is the manuscript presented in an intelligible fashion and written in standard English?

Reviewer #1: Yes

Reviewer #2: No

5. Review Comments to the Author

Reviewer #1: I want to express my heartfelt thanks to the author for their nice endeavor on a very burning issue at lower middle income country context. But i found few confusing parts all over the study. My comments regarding the study is below

1. Authors included both type 1 and type 2 DM. About 290 patients from OPD settings. Patient selection criteria is confusing and not that clear. Line no 157 is so much confusing.

2. How much time is required for interviewing per patients in that OPD settings?

3. At the study, majority of the participants were from rural area and 21.7% were not able to write and read. But analysis is showing that more than half of the participants had good self care knowledge. So, how the authors want to explain that situation as it is not normal. I did not find any proper judgement on discussion regarding this issue.

4.How the authors classified the socio-economic background as poor, medium and rich? There is no explanation on this context. Have they used world bank classification or anything other else scale?

5.Is there any explanation from the authors side , why type 1 DM patients are more than 50% on their study as because it is not normal. We know about 90% of DM are type 2. So, it needs clarification.

6. Age categorization and distribution on analysis showed some irrelevancy. Most of the patients were at that group where they should not be. So, how authors will evaluate this?

7.Analysis showed 15.5% students had DM. What is their type of DM? It should be mentioned.

8. How have the authors evaluated the following situations of diabetic complications?

a. Have they diagnosed retinopathy? What was the retinal evaluation procedure?

b. Have they searched for diabetic neuropathy?

9. Duration of Diagnosis of DM and age grouping of DM is confusing and contradictory.

10. It can be easily described 'why the patients had poor attitude?" . So, normally we can expect who have good knowledge, may have good attitude. This explanation is missing from the study.

Reviewer #2: Please change the title. Knowledge and attitude towards diabetes not self-care practices. in method, it should be described more details. other comments are included in the manuscript. please check. follow those instructions and suggestions.

6. PLOS authors have the option to publish the peer review history of their article (what does this mean?). If published, this will include your full peer review and any attached files.

**Do you want your identity to be public for this peer review?** For information about this choice, including consent withdrawal, please see our Privacy Policy.

Reviewer #1: **Yes: **Fardina Rahman Omi

Reviewer #2: **Yes: **Farzana Saleh

---

## [Decision Letter · Decision Letter 1]

18 Oct 2021

PGPH-D-21-00297R1

Self-Care Knowledge, Attitude and Associated Factors among Outpatients with diabetes mellitus in Arsi Zone, Southeast Ethiopia

Dear Dr. Kassa,

Thank you for submitting your manuscript to PLOS Global Public Health. After careful consideration, we feel that it has merit but does not fully meet PLOS Global Public Health’s publication criteria as it currently stands. Therefore, we invite you to submit a revised version of the manuscript that addresses the points raised during the review process.

Dear Authors, 

Please see editor and reviewers comments below and requesting for addressing the comments accordingly.

We look forward to receiving your revised manuscript.

Kind regards,

Palash Chandra Banik, MPhil

Academic Editor

Journal Requirements:

Additional Editor Comments (if provided):

Dear Authors, I found the tables of the manuscript need to be furnished. Please include the units of the variables like Kg/m2 for BMI, years/ months for duration of diagnosis. Please provide the foot note what the steric (*) indicating. Beside this, I will suggest for professional language editing.

Reviewers' comments:

Reviewer's Responses to Questions

**Comments to the Author**

1. If the authors have adequately addressed your comments raised in a previous round of review and you feel that this manuscript is now acceptable for publication, you may indicate that here to bypass the “Comments to the Author” section, enter your conflict of interest statement in the “Confidential to Editor” section, and submit your "Accept" recommendation.

Reviewer #1: All comments have been addressed

Reviewer #2: All comments have been addressed

2. Does this manuscript meet PLOS Global Public Health’s publication criteria? Is the manuscript technically sound, and do the data support the conclusions? The manuscript must describe methodologically and ethically rigorous research with conclusions that are appropriately drawn based on the data presented.

Reviewer #1: Yes

Reviewer #2: Yes

3. Has the statistical analysis been performed appropriately and rigorously?

Reviewer #1: Yes

Reviewer #2: Yes

4. Have the authors made all data underlying the findings in their manuscript fully available (please refer to the Data Availability Statement at the start of the manuscript PDF file)?

Reviewer #1: Yes

Reviewer #2: Yes

5. Is the manuscript presented in an intelligible fashion and written in standard English?

Reviewer #1: Yes

Reviewer #2: Yes

6. Review Comments to the Author

Reviewer #1: Authors have changed the manuscript rigorously and approached all the comments. The tables are not still well organized. Authors can present them more beautifully by inserting symbolic marks and at end by giving the explanation. Otherwise, whole study is changed drastically and in well condition.

Reviewer #2: Recommendation for submission

7. PLOS authors have the option to publish the peer review history of their article (what does this mean?). If published, this will include your full peer review and any attached files.

**Do you want your identity to be public for this peer review?** For information about this choice, including consent withdrawal, please see our Privacy Policy.

Reviewer #1: **Yes: **Fardina Rahman Omi

Reviewer #2: **Yes: **Dr. Farzana Saleh

---

## [Editor Report · Decision Letter 2]

15 Nov 2021

Self-Care Knowledge, Attitude and Associated Factors among Outpatients with diabetes mellitus in Arsi Zone, Southeast Ethiopia

PGPH-D-21-00297R2

Dear Dr. Kassa,

We're pleased to inform you that your manuscript has been judged scientifically suitable for publication and will be formally accepted for publication once it meets all outstanding technical requirements.

Within one week, you'll receive an e-mail detailing the required amendments. When these have been addressed, you'll receive a formal acceptance letter and your manuscript will be scheduled for publication.

An invoice for payment will follow shortly after the formal acceptance. To ensure an efficient process, please log into Editorial Manager at https://www.editorialmanager.com/pgph/ click the 'Update My Information' link at the top of the page, and double check that your user information is up-to-date. If you have any billing related questions, please contact our Author Billing department directly at authorbilling@plos.org.

Kind regards,

Palash Chandra Banik, MPhil

Academic Editor

Additional Editor Comments (optional): Thank you for addressing all the issues addressed by the reviewers. 